# Expanding understanding of chick embryo's nervous system development at HH22-HH41 embryonic stages using X-ray microcomputed tomography

Igor Rzhepakovsky[1], Sergey Piskov[1], Svetlana Avanesyan[1], Magomed Shakhbanov[1], Marina Sizonenko[1], Lyudmila Timchenko[1], Andrey Nagdalian[1], Mohammad Ali Shariati[2], Ammar Al-Farga[3], Faisal Aqlan[4]*, Andrey Likhovid[1]

1 North-Caucasus Federal University, Stavropol, Russia, 2 Semey Branch of Kazakh Research Institute of Processingand Food Industry, Almaty, Kazakhstan, 3 Department of Biochemistry, College of Sciences, University of Jeddah, Jeddah, Saudi Arabia, 4 Department of Chemistry, College of Sciences, Ibb University, Ibb, Yemen

* aqlanfaisal@gmail.com

**Data Availability Statement:** All data files are available from the authours (attaches in the system).

## Abstract

Assessing the embryotoxicity and teratogenicity of various substances and processes is crucial due to their complexity and resource intensity. The chicken embryo (CE) serves an ideal model for simulating the first months of mammalian embryonic development. This makes the CE a reliable model for testing teratogenic effects, particularly in relation to the nervous system (NS), which experiences developmental abnormalities second in frequency only to cardiovascular teratogenic disorders. Microcomputed tomography (μCT) is a promising method for studying these processes. The advantages of μCT include relatively high research speed, diagnostic accuracy, high resolution and the ability to visualize the entire internal 3D structure of an object while preserving for other types of research. At the same time, there are practically no available databases of normative μCT data, both qualitative and quantitative, which would act as a starting point for screening detection of abnormalities in the development of the NS. In this study, we present a simple method for obtaining very detailed quantitative sets of 2D and 3D μCT data of NS structures of the CE (*Gallus Gallus domesticus*) at HH22-HH41 embryonic stages with contrasting by 1% phosphotungstic acid. The results of μCT demonstrate the exact boundaries, high general and differentiated contrast of the main and specific structures of the NS of CE, which are quantitatively and qualitatively similar to results of histological analysis. Calculations of the X-ray density and volume of the main structures of the NS at constant exponential growth are presented. In addition to the increase in linear dimensions, significant changes in the structures of various parts of the brain were identified and visualized during the CE development at HH22 to HH41 embryonic stages. The data presented establish the first methodology for obtaining normative data, including subtle localized differences in the NS in CE embryogenesis. The data obtained open up new opportunities for modern embryology, teratology, pharmacology and toxicology.

**Funding:** The study was funded by the Russian Science Foundation in the form of a grant [23-24-00282] to IR (https://rscf.ru/en/project/23-24-00282/ - last accessed 19 October 2024).

**Competing interests:** No any competing interests.

## 1. Introduction

The assessment of teratogenicity in new drugs, cosmeceuticals, and food additives remains a highly relevant challenge [1, 2]. Teratogenicity studies are particularly high priority due to their mechanistic complexity and resource-intensive nature. Demonstrating a teratogenic effect is a difficult task that requires the acquisition of experimental teratological data through animal testing [3]. However, due to ethical considerations, as well as efforts to reduce costs and expedite the early stages of teratogenicity screening moving on to preclinical experiments, there is growing scientific interest in alternative effective and cost-efficient model systems. These systems are intended for early screening of pharmacological agents, serving as a precursor to the more extensive standard series of tests [4–7].

In the context of congenital anomalies, the developing chicken embryo (CE) model has garnered significant scientific attention due to its utility in understanding embryonic development across different models [8, 9]. The CE, particularly at early stage of embryogenesis, serves as an ideal model that closely simulates the first months of mammalian development. Notably, the stages of neuronal and vertebral system development in the CE closely resemble those of the human embryo [10]. This similarity makes the CE a highly convenient and reliable model for testing teratogenic effects, particularly concerning the nervous system (NS). NS abnormalities rank second in frequency after cardiovascular teratogenic disorders [11, 12].

Defects in the development of the NS are inherently complex and challenging to detect in model systems, largely due to the continuous spatial structural reorganization of the central nervous system organs throughout embryogenesis [13]. The use of conventional pathomorphological approaches for detailed visualization of the morphology and abnormalities of the NS at various stages of development is limited and can result in confusion regarding the spatial and temporal definition of structures. This limitation is increasingly driving researchers to adopt modern methods of 3D visualization.

For instance, Lazcano et al. [14] performed a 3D reconstruction of the brain of juvenile axolotl (*Ambystoma mexicanum*) using magnetic resonance imaging (MRI). Perelsman et al. [15] improved the expansion microscopy protocol for zebrafish larvae and their volumetric brain assessment. Ambekar et al. [16] obtained and processed structural and mechanical maps of mouse embryos through the synchronized use of the Brillouin microscopy system and optical coherence tomography (OCT). Similarly, Meombe Mbolle et al. [17] conducted structural neuroimaging of mouse embryos using photoacoustic tomography. Ishii et al. [18] carried out 3D observations of morphological changes in the brain of *Xenopus tadpole* using X-ray microtomography (μCT).

Nowadays, novel 3D scanning and image processing systems offer some degree identification of structures of the developing NS of CE using μCT methods [19], MRI [20–22], OCT [23], high-frequency ultrasound imaging [24], and mesoscale selective planar illumination microscopy (mesoSPIM) [25].

The MRI method, despite being one of the most common tools of 3D visualization, still cannot fully reproduce the complexity of CE tissue trajectories and the organization. While MRI and high-frequency ultrasound imaging can generate fully registered 3D image datasets, they are limited by the size of the imaging field, reduced spatiotemporal resolution, and low differential tissue contrast [20, 21, 24]. OCT can produce and visualize 3D images with high spatial resolution; however, visualizing CE at late stages using OCT remains a difficult task due to insufficient penetration depth, restricting its use to early embryogenesis [23]. The spatial resolution capabilities of the mesoSPIM system are promising, but its accessibility remains a limitation for many researchers [25].

μCT is particularly advantageous method due to its high differential contrast, availability of equipment and low cost of scanning [26–28]. However, to identify and characterize

teratogenic defects in the NS in an animal model, it is essential to have in-depth practical morphological knowledge of the normal organ and tissue architecture at each development stage. This is important as the timing of key developmental events may vary significantly among different organs and organisms [13, 29]. Currently, there are almost no available databases of normative µCT data, both in terms of visualization and quantitative metrics, which could act as a baseline for detecting of abnormalities in NS development. Obtaining such data is critically important. It would greatly enhance the potential of the CE as a biological model and demonstrate the capabilities of µCT for screening teratogenic effects, establishing microtomographic criteria, and identifying markers of various embryopathies.

Notably, to date, there are only a few resources for the 3D visualization of secondary neurulation by µCT. Choi S. et al. [19] demonstrated the spatial-temporal relationship of the caudal cell mass with the floor plate of the neural tube and the notochord [19]. The authors described dynamic changes in the chordoneural hinge, cavitation of the secondary neural tube and the primitive strip during the early stages of secondary neurulation. However, it is important to note that the information provided is isolated and focuses primarily on the earliest stages of the NS development. Teratogenic defects in the NS development can occur not only during neurulation, but also at later stages of embryogenesis [30].

The purpose of this work was to illustrate the morphological details of the development of the NS of CE in the dynamics of various stages of embryogenesis using µCT. This study aims to provide, for the first time, an easily accessible resource (online) of microtomograms, as well as reconstructed 2D and 3D images with high resolution, and quantitative µCT indicators of the development of the NS of CE. These resources are critically important for better understanding the normal processes of the NS development and for more objectively assessment the capabilities and diagnostic significance of µCT in screening study for teratogenic reactions.

## 2. Materials and methods

### 2.1. Chemicals

For the experiment, formalin solution, neutral buffered 10%, isopropyl alcohol ≥99.7%, ethyl alcohol 95%, medical paraffin Histomix, hematoxylin and eosin were obtained from Biovitrum (Russia) and phosphotungstic acid hydrate 99.99% was purchased from Sigma-Aldrich (USA).

### 2.2. Embryo preparation

The study was conducted in accordance with the Helsinki Declaration. The design of the experiments was approved by the local Ethics committee of North Caucasus Federal University (Protocol No. 003 dated 03 August 2023).

Fertilized eggs of the Hysex Brown breed, each weighing 50–55 g, were obtained from the commercial hatchery of Agrokormservice Plus (Giaginskaya village, Republic of Adygea, Russia). The eggs were incubated at 37.5°C and 50% relative humidity in a digital incubator Rcom Maru Deluxe Max 380 (Autoelex CO, Gyeongsangnam-do, Korea). The trays were rotated automatically every 2 hours. Each egg was screened daily to confirm viability, and the level of CE development was monitored using the PKYA-10 ovoscope (Premier, Moscow, Russia). The stage of development was determined by the Hamburger and Hamilton (HH) system [31]. CE selection was included 5 samples for each day of development from 4[th] (HH 22–24) to 15[th] (HH41) days.

According to the recommendations for euthanasia [32, 33], the CE were euthanized by cooling at 4°C for 4 hours at HH22-24 –HH33-34 stages and by a 30 minutes $CO_2$ (70%) exposure at HH36 –HH41 stages. CE were extracted by removing a portion of the eggshell and inner membrane at the blunt end of the egg, above the air chamber, with the edge previously

outlined during ovoscopy. The extracted CE were carefully washed with saline solution and fixed in neutral buffered formalin for 72 hours. Structurally abnormal CE or mechanically injured ones were excluded from the study.

Macroscopic examination of CE selected at different days of incubation was conducted using an Axio Zoom light microscope (Carl Zeiss Microscopy, Germany) at various magnifications. Image capture was performed with a specialized AxioCam MRc5 camera (Carl Zeiss Microscopy, Germany) and Zen 2 Pro software (Carl Zeiss Microscopy, Germany).

## 2.3. Stain

Phosphotungstic acid (1% PTA) was used as contrast stain for CE (4–15 day, HH22-HH41). The CE (4–8 day, HH22-HH34) fixed in a 10% buffered formalin solution for 72 hours were washed under running water for 12 hours, dehydrated in replaceable portions of ethanol 30% (2 hours), 50% (2 hours), 70% (12 hours) and placed in solution of radiopaque stain at 1:20 (V of CE to V of solution), and were kept at 40˚C for 24 hours. The CE (9–15 day, HH35-HH41) fixed in a 10% buffered formalin solution for 96 hours were washed under running water for 24 hours, dehydrated in replaceable portions of ethanol of 30% (2 hours), 50% (2 hours), 70% (12 hours) and placed in solution of radiopaque stain 1% PTA at 1:20 (volume of CE to volume of solution) for 96 hours [34].

## 2.4. μCT imaging systems

For μCT, different tubes were used based on the age of the chicken embryos (CE): Eppendorf Safe-Lock Tubes 2 mL colorless (polypropylene) for 4–7 days CE (HH22-HH32), Servicebio Centrifuge Tubes 15 mL colorless (polypropylene) for 8–12 days CE (HH33-HH38), and Servicebio Centrifuge Tubes BioBased 50 mL colorless (polypropylene) for 13–15 days CE (HH39-HH41). Test tubes containing the CE samples in 70% ethanol solution were transferred to the Skyscan 1176 microtomograph (Bruker, Kontich, Belgium) and fixed in place with foam retainers. Notably, the use of 70% ethanol solution is justified by low radiopacity, allowing clear visualization of even the lowest contrast parts of the object [28, 35].

CE scanning was performed by rotating an 11-megapixel camera (4000×2672 pixels) by 180˚ (0.3˚/step), with averaging of three images per step. This resulted in an isometric spatial resolution 8.87 μm (for 4–12 day, HH22-HH38) and 17.74 μm (for 13–15 day, HH39-HH41). The radiation passing level in most planes of the CE was maintained between 30–50%. When reconstructing the histogram (grey value) of all images was included in the contrast region (minimum and maximum gray values) to most effectively highlight the general and differentiated contrast of parts of the object. A wider filter indicated an increase in the overall level of object contrast, helping to distinguish the embryo from the surrounding space [34].

Grouped images (image stacks) were processed and reconstructed into 3D data sets using NRecon 1.7.4.2 software (Bruker, Kontich, Belgium). The process took about 2 hours for each sample tube. Post-processing, alignment, orientation in space (x, y, z), mapping of X-ray contrast profiles and highlighting of individual areas of reconstructed materials were carried out using DataViewer 1.5.6.2 software (Bruker, Kontich, Belgium). Visualization of 3D images was carried out in the CTvox 3.3.0r1403 software (Bruker, Kontich, Belgium). Morphometry and evaluation of the X-ray density of various CE structures in Hounsfield units (HU) were carried out using CT-analyser 1.18.4.0 software (Bruker, Kontich, Belgium) according to our own methods described in previous works [36, 37]. μCT scan settings used in the experiment are presented in Table 1.

Volume segmentation of 3D images was performed using the algorithm recommended by Bruker-microCT (Kontich, Belgium). A consistent allocation of areas of interest was carried

out. The resulting structures were saved as separate volumes. For most of the quantitative calculations present in this article, the organ or region of interest was segmented by creating negative space (i.e., setting pixel values in that region to zero contrast) and propagating that negative space through the Z plane. This method ensured only one tissue type was captured, allowing for the independent quantification of tissue or fluid volume in a particular organ. Multiple labeled volumes were derived from each CE and quantified in the same way. Data from at least five embryos were used for each region/tissue.

## 2.5. Histological preparation

To assess the structure of the CE brain on µCT images, following the example of other researchers [31, 38–42], the images were verified with the corresponding histological sections. CE heads taken on days 4–15 of incubation were fixed, washed, dehydrated in isopropyl alcohol, and embedded in medical paraffin Histomix (Biovitrum, St. Petersburg, Russia). Histological sections, 6 µm thick, were prepared using a rotary microtome NM 325 (Thermo Fisher Scientific, Waltham, US). The histological sections were then stained with hematoxylin and eosin follwoing generally accepted protocols [43].

Briefly, the histological sections were dewaxed by incubation in xylene (2 cycles), dehydrated in ethanol at concentrations of 95%, 80% and 70%, and washed with distilled water. Then, the histological sections were incubated in hematoxylin solution (3 minutes), washed in tap water, incubated in 1% aqueous eosin solution (5 minutes) and washed with distilled water. After removing the water spills, the histological sections were incubated in 96% ethanol and xylene and closed in a mounting medium Vitrogel (Biovitrum, St. Petersburg, Russia).

Photos of micropreparations were obtained using Axio Zoom V16 and Axio Imager 2 (A2) research class microscopes (Carl Zeiss Microscopy, Oberkochen, Germany). These images were captured with a specialized AxioCam MRc5 camera and Zen 2 Pro software (Carl Zeiss Microscopy, Oberkochen, Germany).

## 2.6. Statistical data processing

For each embryonic stages indicated in Tables 1 and 2, 5 samples were used. The most characteristic representative materials were selected for visualization of 2D and 3D structures, as well as for visualization of radiopacity profiles. Materials with mechanical and digital defects, which could have resulted from staining or scanning, were excluded from the research. Segmentation and quantification were carried out according to the recommendations of Kim et al. [44]. Skyscan 1176 (software platform Bruker, Kontich, Belgium) running on a Windows 7 Professional (Microsoft Corp., Redmond, WA, USA) workstation with 32 Gb of RAM and an Nvidia Quadro K 4000 graphics card (Nvidia Corp., Santa Clara, CA, USA) was used for µCT data processing.

Individual differences in the samples were assessed using statistical analysis using ANOVA, followed by Tukey post hoc testing with $p < 0.05$ as a significance threshold.

## 3. Results and discussion

The data obtained from the µCT analysis of X-ray density and the visualized volume of the CE, brain, eye (left) and spine are presented in Table 2.

The total X-ray density of CE, the left eye, and the spine decreased during growth and development at 4–8 days (HH22-HH34). This decrease is attributed to both the growth feature and the selective fixation and low penetrating ability of PTA molecules in tissues. Specifically, the total X-ray density decreased by twofold for the CE and brain, 2.5-fold for the spine, and 4.5-fold for the left eye by the HH34 embryonic stage. The more significant decrease in the X-

**Table 1. μCT scan settings for contrast stain of the CE, n = 5.**

| Embryonic stages | Exposure time, h | Tempe-rature,° | Filter | X-ray voltage, kV | X-ray current, μA | Rotation step, deg | Scans averaged | Voxel size (μm) | Smoothing Gaussian | Ring artifact correction | Beam hardening correction (%) | Minimum for CS to image conversion | Maximum for CS to image conversion |
|---|---|---|---|---|---|---|---|---|---|---|---|---|---|
| 4th day, HH22-HH24 | 24 | 40 | Al 0.5 mm | 50 | 500 | 0.3 | 3 | 8.87 | 0 | 10 | 41 | 0.001 | 0.17 |
| 5th day, HH25-HH27 | 24 | 40 | Al 1 mm | 65 | 480 | 0.3 | 3 | 8.87 | 0 | 10 | 41 | 0.001 | 0.1 |
| 6th day, HH28-HH29 | 24 | 40 | Al 1 mm | 65 | 480 | 0.3 | 3 | 8.87 | 0 | 10 | 41 | 0.002 | 0.095 |
| 7th day, HH30-HH32 | 24 | 40 | Al 1 mm | 65 | 480 | 0.3 | 2 | 8.87 | 2 | 10 | 41 | 0.001 | 0.1 |
| 8th day, HH33-HH34 | 24 | 40 | Cu-Al | 80 | 300 | 0.3 | 2 | 8.87 | 0 | 10 | 41 | 0 | 0.07 |
| 9th day, HH35 | 96 | 40 | Cu-Al | 80 | 300 | 0.3 | 2 | 8.87 | 2 | 10 | 41 | 0.001 | 0.09 |
| 10th day, HH36 | 96 | 40 | Cu-Al | 80 | 300 | 0.3 | 2 | 8.87 | 2 | 10 | 41 | 0.001 | 0.085 |
| 11th day, HH37 | 96 | 40 | Cu-Al | 80 | 300 | 0.3 | 2 | 8.87 | 0 | 10 | 41 | 0.002 | 0.095 |
| 12th day, HH38 | 96 | 40 | Cu-Al | 80 | 300 | 0.3 | 2 | 8.87 | 1 | 10 | 41 | 0.002 | 0.09 |
| 13th day, HH39 | 96 | 40 | Cu 0.1 mm | 90 | 270 | 0.3 | 2 | 17.74 | 1 | 10 | 41 | 0 | 0.075 |
| 14th day, HH40 | 96 | 40 | Cu 0.1 mm | 90 | 270 | 0.3 | 2 | 17.74 | 1 | 15 | 41 | 0.002 | 0.06 |
| 15th day, HH41 | 96 | 40 | Cu 0.1 mm | 90 | 270 | 0.3 | 2 | 17.74 | 1 | 10 | 41 | 0 | 0.06 |

**Table 2. X-ray density and visualized volume of the NS of CE at HH22-HH41 embryonic stages, n = 5.**

| Embryonic stages | Chick embryo | | Brain | | Eye (left) | | Spine | |
|---|---|---|---|---|---|---|---|---|
| | X-ray density, HU | visualized volume, mm³ | X-ray density, HU | visualized volume, mm³ | X-ray density, HU | visualized volume, mm³ | X-ray density, HU | visualized volume, mm³ |
| 4th day, HH22-HH24 | 3115.0±111 | 17.7±2 | 5284.0±202 | 3.7±0.2 | 7514.0±257 | 0.31±0.04 | 6978.0±320 | 1.58±0.1 |
| 5th day, HH25-HH27 | 2065.0±116 | 105.3±5 | 2396.0±148 | 22.3±2 | 3780.0±180 | 3.94±0.5 | 5745.0±212 | 6.7±0.4 |
| 6th day, HH28-HH29 | 2447.0±125 | 137.6±9 | 2753.0±112 | 27.9±2 | 3255.0±177 | 6.54±0.5 | 6339.0±369 | 12.5±0.5 |
| 7th day, HH30-HH32 | 1517.0±84 | 367.6±23 | 2394.0±135 | 54.5±3 | 1693.0±95 | 33.3±2 | 2659.0±139 | 37.6±3.2 |
| 8th day, HH33-HH34 | 1807.0±103 | 571.3±38 | 3238.0±193 | 80.8±5 | 1897.0±98 | 66.0±3 | 2775.0±173 | 51.0±4.8 |
| 9th day, HH35 | 7439.0±302 | 762.0±45 | 8613.0±324 | 84.6±6 | 4730±202 | 85.8±9 | 6816.0±304 | 83.2±8 |
| 10th day, HH36 | 7312.0±248 | 950.0±53 | 10052.0±466 | 73.6±5 | 4112±260* | 114.8±14* | 8638.0±478 | 102.4±12 |
| 11th day, HH37 | 6960.0±212 | 1269.0±91 | 9092.0±503 | 102.3±9 | 4832±245* | 127.7±12* | 7072.0±406 | 139.8±11 |
| 12th day, HH38 | 6759.0±135 | 2101.0±184 | 7344.0±324 | 186.1±29 | 3751±209* | 188.7±26* | 8410.0±398 | 240.4±33 |
| 13th day, HH39 | 6310.0±289 | 3123.7±195 | 6705.0±270 | 260±33 | 14234±544* | 100.0±21* | 7379.0±441 | 348.0±31 |
| 14th day, HH40 | 6819.0±315 | 5824.2±214 | 9971.0±389 | 454±38 | 5449±286* | 358.0±29* | 8753.0±475 | 494.0±39 |
| 15th day, HH41 | 7151.0±332 | 6476.0±245 | 10482.0±401 | 560±44 | 6416±345* | 313.6±24* | 9233.0±491 | 569.1±43 |

*a significant proportion of the CE showing deformation of the eyeball

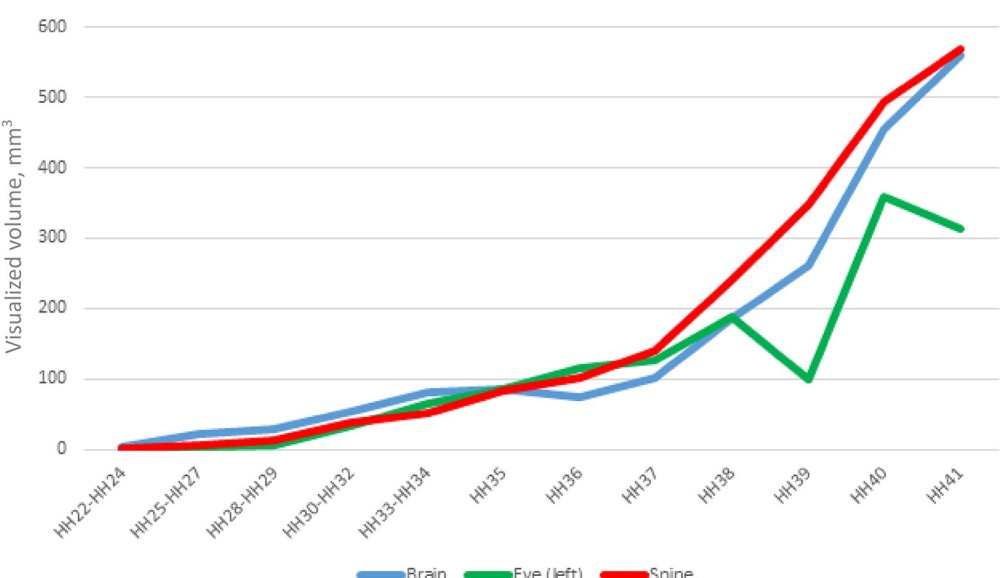

**Fig 1. Dynamics of the visualized volume of nervous system of chick embryo at HH22-HH41 embryonic stages, n = 5.**

ray density of the eyes is likely due to a significant increase in the cavity of the eyes [45, 46]. Concurrently, the X-ray density of the brain varies unevenly, which is associated with an increase in the volume of nervous tissue well-contrasted by PTA and a decrease in the volume of the cavities of the cerebral bladders or ventricles during the development of CE [20].

However, from 9–15 days (HH35-HH41), with an increase in staining exposure to 96 hours, a significant increase in the X-ray density of all studied tissues was observed. This high level of radiopacity was maintained through the HH41 embryonic stage. The X-ray density measurements were as follows: CE ranged from 6310 to 7439 HU, the brain from 6705 to 10482 HU, the left eye from 3751 to 14234 HU, and the spine from 6816 to 9233 HU. A high level of differentiated contrast among the studied tissues was visualized at all studied embryonic stages.

Analysis of the visualized volume of embryos during incubation reveals the following peaks in growth intensity: a sixfold increase at 4–5 days, a 2.5-fold increase at 6–7 days, and a twofold increase at 13–14 days. At other embryonic stages studied, the visualized volume of CE increased by 10 to 65%.

Fig 1 illustrates the dynamics of changes in the visualized volume of the main structures related to the NS. The selected structures of the brain, eyes (left) and spine show comparable visualized volume and increase evenly during embryogenesis. Notably, growth and development of the chicken embryo (CE) intensify from the HH36 embryonic stage.

The significant growth and morphological changes of the eye, starting from the HH36 embryonic stage, lead to frequent deformation of the eyeball, which is reflected in the calculations of the total X-ray density and the visualized volume. This observation is supported by other researchers and must be considered in the analysis [47, 48]. However, certain functionally important elements, such as various membranes, pupil, lens, cornea, retina and intraocular optic nerve, remain unchanged at the tissue and cellular levels, confirming the results of the histological analysis carried out by Kim et al. [44].

The next stage of the work involved visualizing and identifying both macro- and microstructures of the brain during μCT analysis of the CE head (4–15 days, HH22-HH41). Figs 2–6

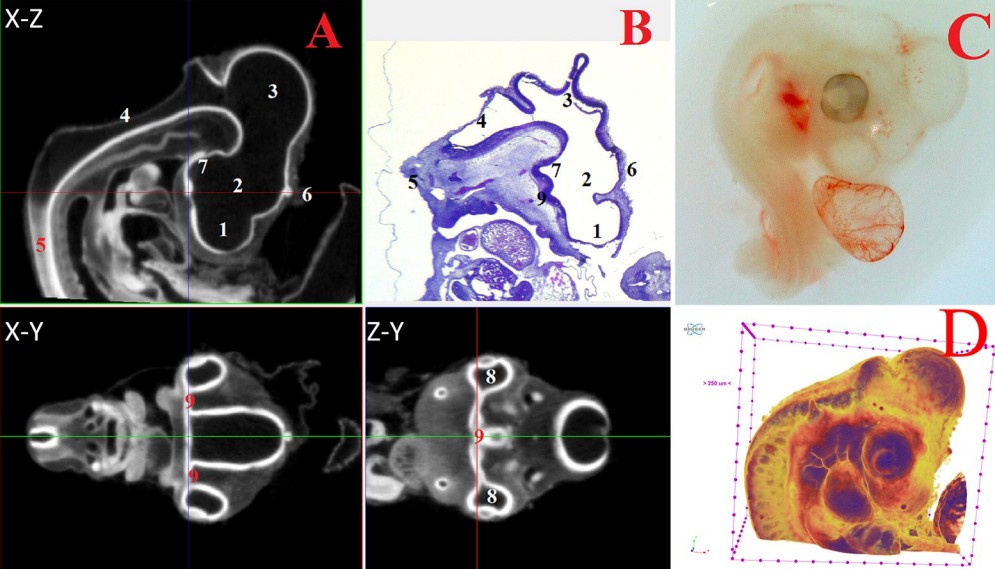

**Fig 2.** Representative cross-sectional images of the head region of a chick embryo at day 4 (embryonic stages HH22-24), counterstained with 1% PTA for 24 h at 40°C (A): coronal (X-Z), transaxial (X-Y) and sagittal (Z-Y) planes, histological section with hematoxylin-eosin staining at magnification ×18.0 (B), still image embryo under the stereoscope (C) and isosurface 3D renderings of the head region (D). The following structures are marked on the images: 1 –telencephalon (lat. *telencephalon*); 2 –diencephalon (lat. *diencephalon*); 3 –midbrain (lat. *mesencephalon*); 4 –hindbrain (lat. *rhombencephalon*); 5 –spinal cord (lat. *medulla spinalis*); 6 –pineal gland (lat. *corpus pineale*); 7 – pituitary gland (lat. *hypophysis*); 8 –eyes (lat. *oculus*); 9 –chiasma and optic nerves (lat. *chiasma opticum*).

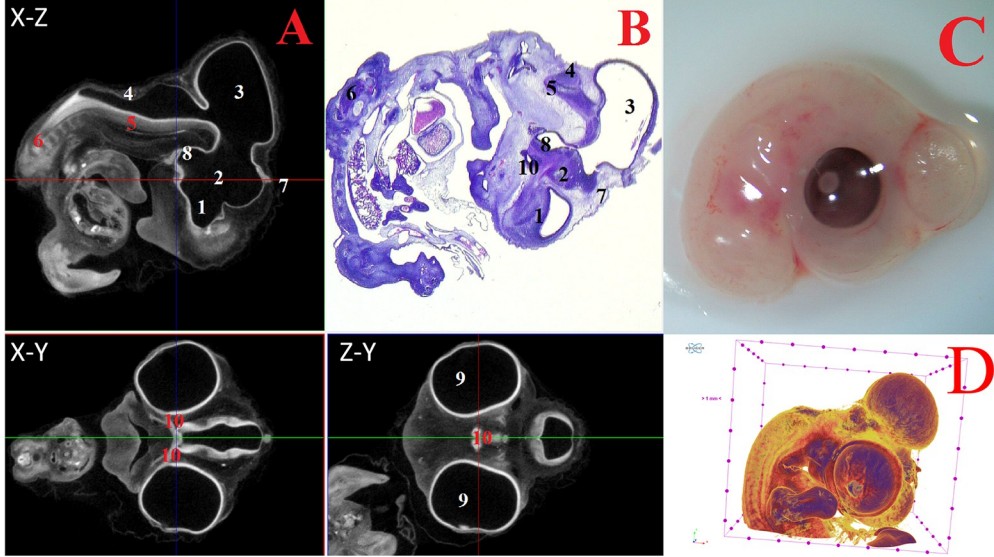

**Fig 3.** Representative cross-sectional images of the head region of a chick embryo at day 6 (embryonic stages HH28-29), counterstained with 1% PTA for 24 h at 40°C (A): coronal (X-Z), transaxial (X-Y) and sagittal (Z-Y) planes, histological sagittal section with hematoxylin-eosin staining at magnification ×13.0 (B), still image embryo under the stereoscope (C) and isosurface 3D renderings of the head region (D). The following structures are marked on the images: 1 –telencephalon (lat. *telencephalon*); 2 –diencephalon (lat. *diencephalon*); 3 –midbrain (lat. *mesencephalon*); 4 –hindbrain (lat. *rhombencephalon*); 5 –medulla oblongata (lat. *myelencephalon*); 6 –spinal cord (lat. *medulla spinalis*); 7 –pineal gland (lat. *corpus pineale*); 8 –pituitary gland (lat. *hypophysis*); 9 –eyes (lat. *oculus*); 10 –chiasma and optic nerves (lat. *chiasma opticum*).

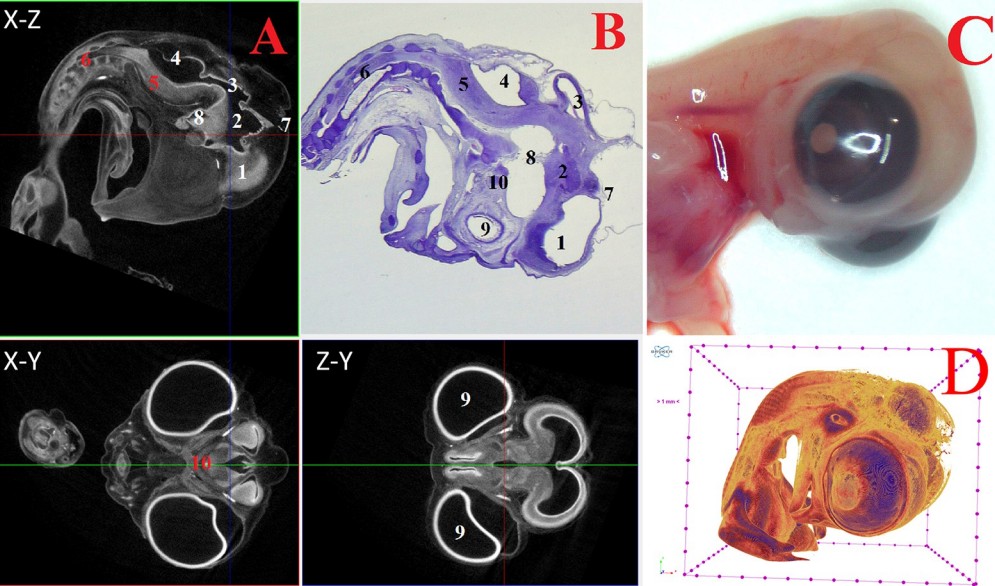

**Fig 4.** Representative cross-sectional images of the head region of a chick embryo at day 8 (embryonic stages HH33-34), counterstained with 1% PTA for 24 h at 40°C (A): coronal (X-Z), transaxial (X-Y) and sagittal (Z-Y) planes, histological sagittal section with hematoxylin-eosin staining at magnification ×13.0 (B) and still image embryo under the stereoscope (C) and isosurface 3D renderings of the head region (D). The following structures are marked on the images: 1 –telencephalon (lat. *telencephalon*); 2 –diencephalon (lat. *diencephalon*); 3 –midbrain (lat. *mesencephalon*); 4 –hindbrain (lat. *rhombencephalon*); 5 –medulla oblongata (lat. *myelencephalon*); 6 –spinal cord (lat. *medulla spinalis*); 7 –pineal gland (lat. *corpus pineale*); 8 –pituitary gland (lat. *hypophysis*); 9 –eyes (lat. *oculus*); 10 –chiasma and optic nerves (lat. *chiasma opticum*).

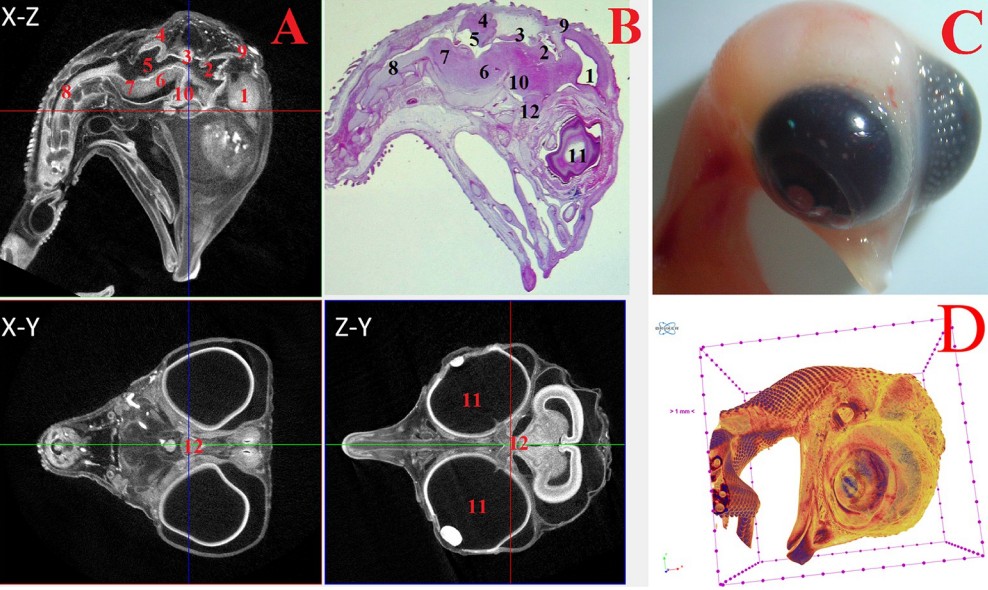

**Fig 5.** Representative cross-sectional images of the head region of a chick embryo at day 11 (embryonic stages HH37), counterstained with 1% PTA for 96 h at 40°C (A): coronal (X-Z), transaxial (X-Y) and sagittal (Z-Y) planes, histological sagittal section with hematoxylin-eosin staining at magnification ×8.5 (B), still image embryo under the stereoscope (C) and isosurface 3D renderings of the head region (D). The following structures are marked on the images: 1 –telencephalon (lat. *telencephalon*); 2 –diencephalon (lat. *diencephalon*); 3 –midbrain (lat. *mesencephalon*); 4 –cerebellum (lat. *cerebellum*); 5 –fourth ventricle (lat. *ventriculus quartus*); 6 –pons (lat. *pons*); 7 –medulla oblongata (lat. *myelencephalon*); 8 –spinal cord (lat. *medulla spinalis*); 9 –pineal gland (lat. *corpus pineale*); 10 –pituitary gland (lat. *hypophysis*); 11 –eyes (lat. *oculus*); 12 –chiasma and optic nerves (lat. *chiasma opticum*).

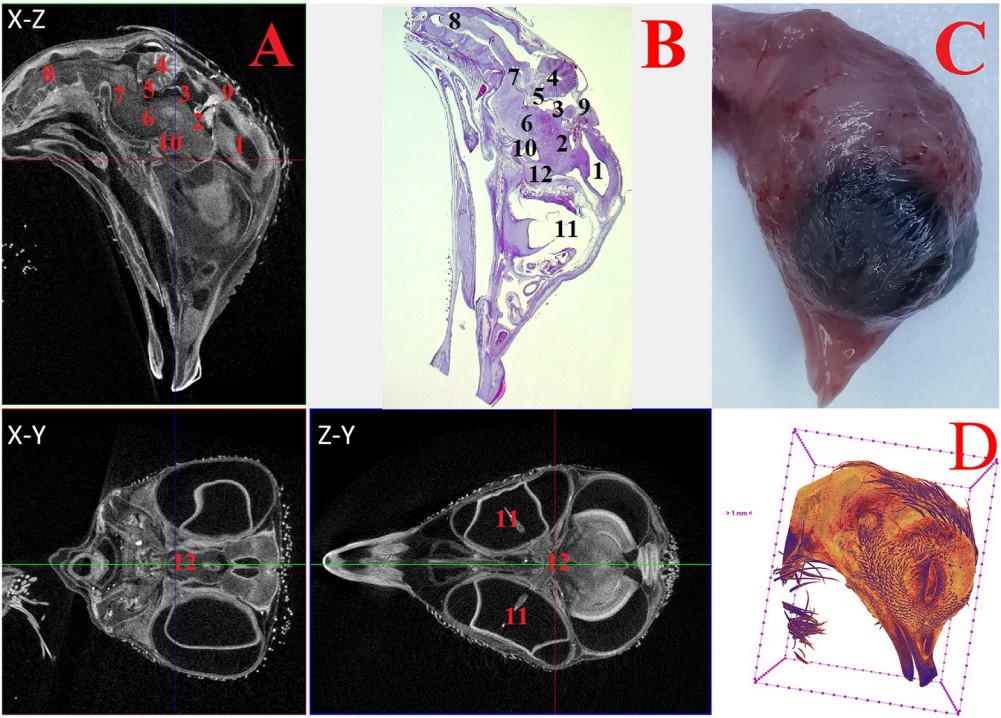

**Fig 6.** Representative cross-sectional images of the head region of a chick embryo at day 14 (embryonic stages HH40), counterstained with 1% PTA for 96 h at 40°C (A): coronal (X-Z), transaxial (X-Y) and sagittal (Z-Y) planes, histological sagittal section with hematoxylin-eosin staining at magnification ×8.0 (B), still image embryo under the stereoscope (C) and isosurface 3D renderings of the head region (D). The following structures are marked on the images: 1 –telencephalon (lat. *telencephalon*); 2 –diencephalon (lat. *diencephalon*); 3 –midbrain (lat. *mesencephalon*); 4 –cerebellum (lat. *cerebellum*); 5 –fourth ventricle (lat. *ventriculus quartus*); 6 –pons (lat. *pons*); 7 –medulla oblongata (lat. *myelencephalon*); 8 –spinal cord (lat. *medulla spinalis*); 9 –pineal gland (lat. *corpus pineale*); 10 –pituitary gland (lat. *hypophysis*); 11 –eyes (lat. *oculus*); 12 –chiasma and optic nerves (lat. *chiasma opticum*).

and Supplementary (S1–S7 Figs) demonstrate for the first time, high-resolution microtomographic images of the brain structures, which were verified by parallel histological analysis. The developed contrast technique and selected scanning parameters achieved a high level of general and differential radiopacity in both the surrounding tissues and in individual brain structures at all studied stages of embryonic development.

On the 4th day of incubation (HH22-HH24), the following large structures were identified and well visualized in various projections during μCT analysis: telencephalon (*telencephalon*), diencephalon (*diencephalon*), midbrain (*mesencephalon*), hindbrain (*rhombencephalon*), spinal cord (*medulla spinalis*), pineal gland (*corpus pineale*), pituitary gland (*hypophysis*), eyes (*oculus*), chiasma and optic nerves (*chiasma opticum*) (Fig 2).

At 5–8 days (HH25-HH34), in addition to the structures mentioned above, the medulla oblongata (*myelencephalon*) structures were distinguished. Intensive expansion of the thalamus and telencephalon structures, as well as the general growth of nerve tissues of all parts of the brain were observed at HH25-HH34 embryonic stages (Figs 3 and 4).

At 9–15 days (HH35-HH41), the final formation of structures was observed, including the division of hindbrain into distinct components such as the cerebellum, pons, medulla oblongata. Additionally, the structures of the telencephalon expanded significantly, leading to the formation of the nerve tissues in the cerebral hemispheres (*hemisphaerium cerebri*) (Figs 5 and 6).

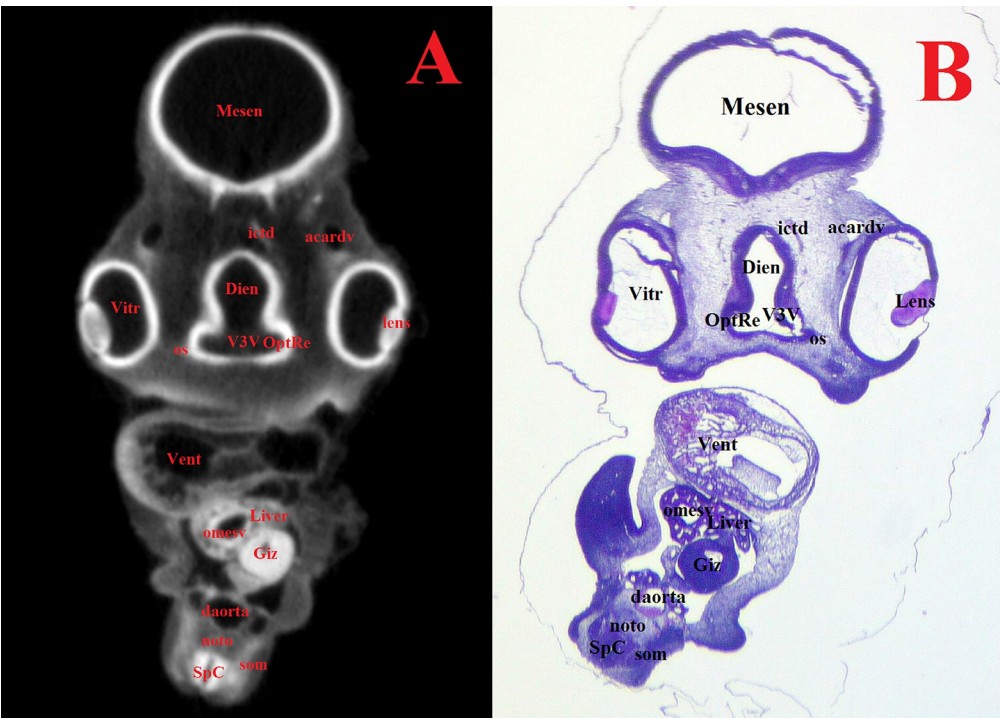

**Fig 7.** Representative frontal sectional images of the head region of a chick embryo, counterstained with 1% PTA for 24 h at 40˚C (A) and histological frontal section with hematoxylin-eosin staining (B, magnification ×18.0) for comparison (day 4, embryonic stages HH22-24). The following structures are marked on the images: Acardv–anterior cardinal vein; Daorta–dorsal aorta; Dien–diencephalon; Giz–gizzard; Ictd–internal carotid artery; Lens–lens; Liver–liver; Mesen–mesencephalon; Noto–notochord; Omesv–omphalomesenteric (vitelline) vein; OptRe–optic recess of 3$^{rd}$ ventricle; Os–optic stalk; Som–somite; SpC–spinal cord; V3V –ventral third ventricle; Vent–ventricle of heart; Vitr–vitreous humor of eye.

Figs 7–11 and Supplementary (S8–S19 Figs) show representative µCT slices of the CE brain at 4 to 15 (HH22–HH41) in various views. The morphology of the CE brain in our µCT model closely matches that observed in histological sections. Analysis of multiple frontal, horizontal and sagittal µCT slices of the CE brain, combined with high-quality differential contrast of specific brain areas, confirmed the ease of orientation and symmetrical alignment of structures in comparison with histological sections.

The Orientation of the studied brain structures was carried out following the recommendations of Hoops et al. [49]. Currently, accurate identifying the localization of the ventricles of the brain and interventricular septa is an important task for determining the brain's macro-structures [20, 50].

However, this did not cause any particular difficulties in the µCT analysis due to low X-ray contrast of the studied structures compared to the surrounding tissues. The identification of specific sites related to the NS was carried out through careful comparison with histological sections, micrographs from electronic atlases and relevant scientific publications. The membranes of the brain and spinal cord, as well as structures such as the pituitary gland, epiphysis, ganglia, nerve trunks, and optic chiasma, exhibited a high level of X-ray contrast and were clearly visualized in 2D and 3D imaging at all studied stages of embryogenesis. Blood vessels were visualized based on contrast difference and appeared as dark, low-contrast formations.

As a result, we identified and marked localization of more than 40 structures related to the NS on the frontal, horizontal and sagittal µCT slices at all studied stages of embryogenesis

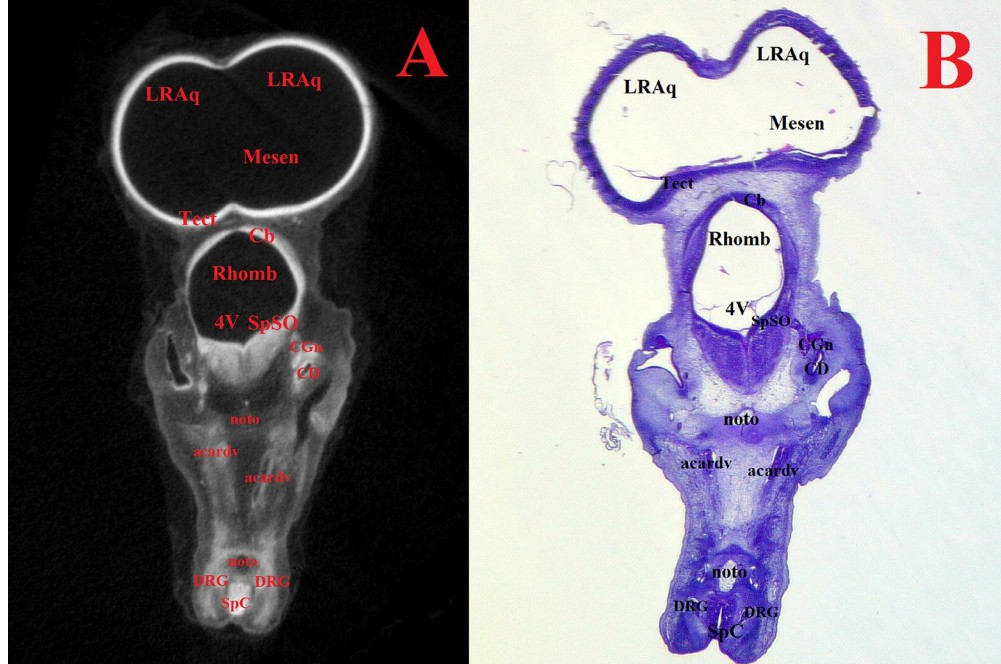

**Fig 8.** Representative frontal sectional images of the head region of a chick embryo, counterstained with 1% PTA for 24 h at 40˚C (A) and histological frontal section with hematoxylin-eosin staining (B, magnification ×17.5) for comparison (day 6, embryonic stages HH28-29). The following structures are marked on the images: 4V –fourth ventricle; Acardv–anterior cardinal vein; Cb–cerebellum; CD–cochlear duct; CGn–cochlear ganglion; DRG–dorsal root ganglion; LRAq–lateral recess of the cerebral aqueduct; Mesen–mesencephalon; Noto–notochord; Rhomb–rhombencephalon; SpC–spinal cord; SpSO–nucleus of trigeminal spinal tr., oral part; Tect–tectum.

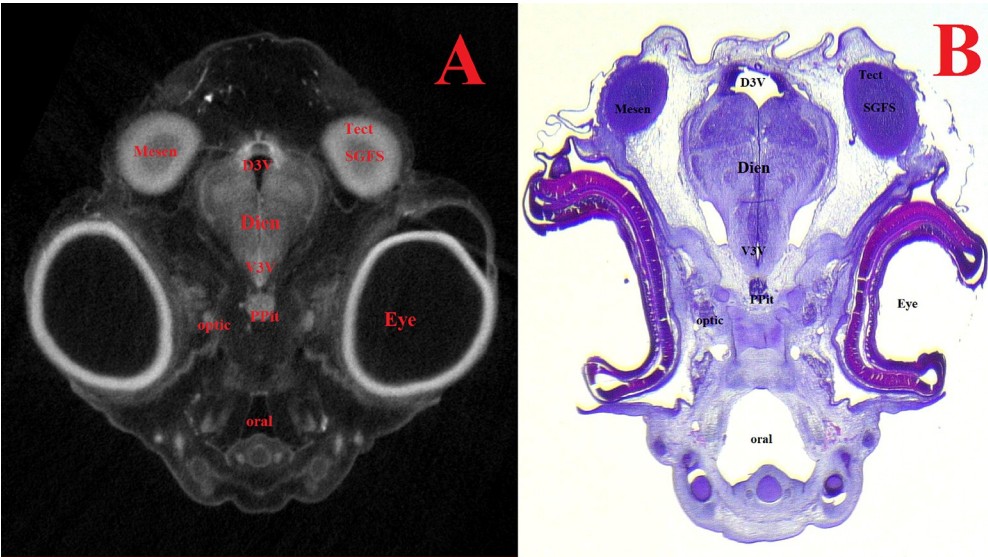

**Fig 9.** Representative frontal sectional images of the head region of a chick embryo, counterstained with 1% PTA for 24 h at 40˚C (A) and histological frontal section with hematoxylin-eosin staining (B, magnification ×13.0) for comparison (day 8, embryonic stages HH33-34). The following structures are marked on the images: D3V –dorsal third ventricle; Dien–diencephalon; Mesen–mesencephalon; Optic–optic nerve; Oral–oral cavity; Ppit–posterior lobe of pituitary gland; SGFS–stratum griseum and fibrosum of tectum; SpC–spinal cord; Tect–tectum; V3V –ventral third ventricle.

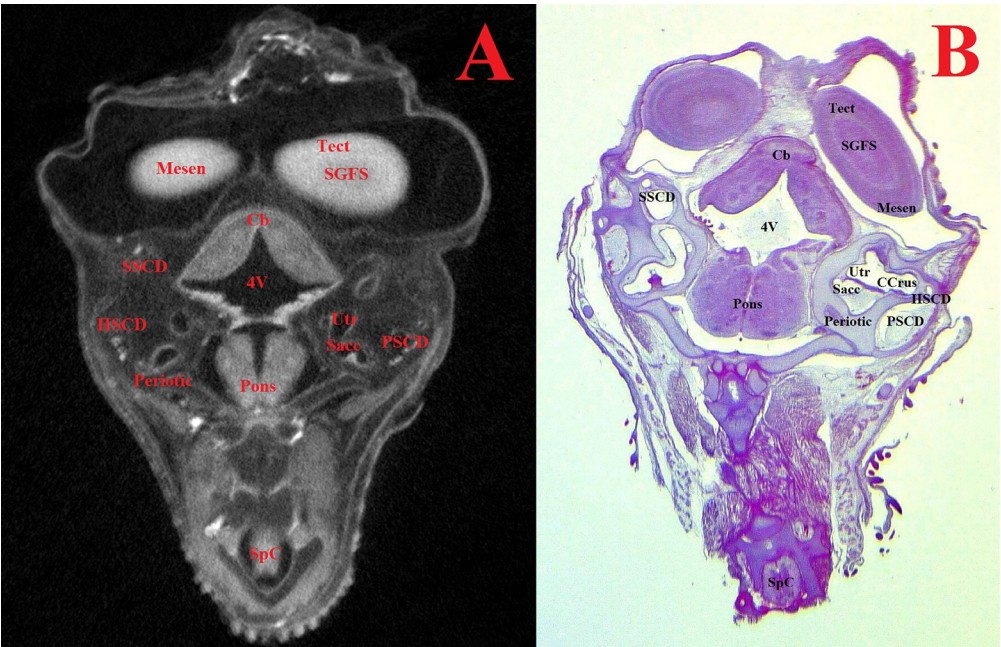

**Fig 10.** Representative frontal sectional images of the head region of a chick embryo, counterstained with 1% PTA for 96 h at 40˚C (A) and histological frontal section with hematoxylin-eosin staining (B, magnification ×10.0) for comparison (day 11, embryonic stage HH37). The following structures are marked on the images: 4V –fourth ventricle; CCrus–common crus of semicircular duct; Cb–cerebellum; HSCD–horizontal semicircular duct; Mesen–mesencephalon; Periotic–periotic capsule; Pons–pons; PSCD–posterior semicircular duct; Sacc–saccule of inner ear; SGFS–stratum griseum and fibrosum of tectum; SpC–spinal cord; SSCD–superior semicircular duct; Tect–tectum; Utr–utricle of inner ear.

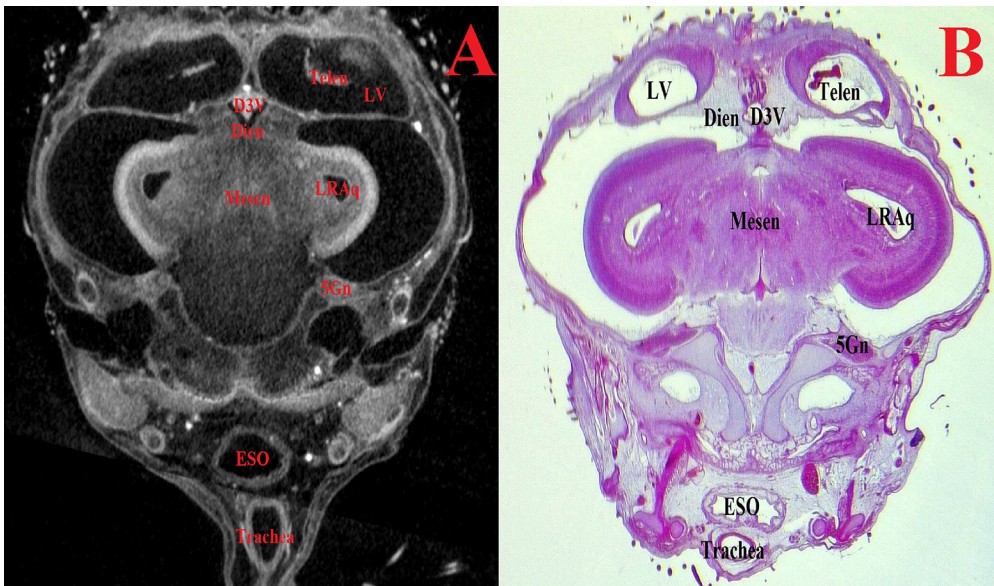

**Fig 11.** Representative frontal sectional images of the head region of a chick embryo, counterstained with 1% PTA for 96 h at 40˚C (A) and histological frontal section with hematoxylin-eosin staining (B, magnification ×10.0) for comparison (day 14, embryonic stage HH41). The following structures are marked on the images: 5Gn–trigeminal ganglion; D3V –dorsal third ventricle; Dien–diencephalon; Eso–esophagus; LRAq–lateral recess of the cerebral aqueduct; LV–lateral ventricle; Mesen–mesencephalon; Telen–telencephalon; Trachea–trachea.

(embryonic stages HH22–HH41). In addition, we designated specific structures such as notochord, dorsal aorta, somite, oral cavity, pharyngeal arch, tongue, etc. This groundwork will pave the way for creating the a new detailed μCT atlas of the developing chicken embryo.

It is worth noting that the obtained results open up significant prospects and, with some adaptation, can become the basis for researchers around the world for both fundamental and applied research on embryonic development.

## 4. Conclusions

This study presents the calculated and visual results of 2D and 3D μCT analysis of the NS structures of the CE (*Gallus Gallus domesticus*) at HH22-HH41 embryonic stages.

Despite the significant increase in the linear dimensions of the chicken embryo (CE) (from 17.7 mm$^3$ to 6476 mm$^3$) and the structures of the nervous system (NS) at the studied embryonic stages (HH22 to HH41), we successfully developed detailed methodological techniques for contrasting, scanning, and processing materials. Additionally, we obtained the most accurate projections for the first time, which is critically important for the reliable analysis of possible changes during embryogenesis.

The results of our μCT demonstrate the exact boundaries and high general and differentiated contrast of the main and specific structures of the NS of CE. These are quantitatively similar to those obtained through histological analysis. In addition to observing an increase in linear dimensions, we identified and visualized changes in the structures of various parts of the brain during the development of HH22-HH41 embryonic stages. The analysis of multiple frontal, horizontal and sagittal μCT slices of the CE brain, combined with high-quality differential contrast of specific areas of the brain areas, confirmed the convenience of orientation and symmetrical alignment of structures when compared histological sections. These results demonstrate that μCT is an effective method of quantitative visualization of the NS.

This study establishes a methodology for obtaining normative data, including subtle localized differences of the NS during embryogenesis. This advancement opens up new opportunities for modern embryology, teratology, pharmacology and toxicology.

### Limitations of the study

In this work, we conducted a μCT analysis of the nervous system of the CE at HH22-HH41 embryonic stages. We found it inappropriate to study the early stages of embryogenesis using this technique due to the insufficient detail of microtomograms. Additionally, μCT analysis of the CE at HH42-HH46 embryonic stages requires improved approaches to contrast, which is necessary due to the large size of the object and the pronounced development of the skin and its derivatives.

### Supporting information

**S1 Fig.** Representative cross-sectional images of the head region of a chick embryo (day 5, embryonic stages HH25-27), counterstained with 1% PTA for 24 h at 40˚C (A): coronal (X-Z), transaxial (X-Y) and sagittal (Z-Y) planes and isosurface 3D renderings of the head region (B). The following structures are marked on the images: 1 –telencephalon (lat. *telencephalon*); 2 – diencephalon (lat. *diencephalon*); 3 –midbrain (lat. *mesencephalon*); 4 –hindbrain (lat. *rhombencephalon*); 5 –medulla oblongata (lat. *myelencephalon*); 6 –spinal cord (lat. *medulla spinalis*); 7 –pineal gland (lat. *corpus pineale*); 8 –pituitary gland (lat. *hypophysis*); 9 –eyes (lat. *oculus*); 10 –chiasma and optic nerves (lat. *chiasma opticum*). Visualization of structures in the DataViewer software and CTvox software.
(TIF)

**S2 Fig.**  Representative cross-sectional images of the head region of a chick embryo (day 7, embryonic stages HH30-32), counterstained with 1% PTA for 24 h at 40°C (A): coronal (X-Z), transaxial (X-Y) and sagittal (Z-Y) planes and isosurface 3D renderings of the head region (B). The following structures are marked on the images: 1 –telencephalon (lat. *telencephalon*); 2 –diencephalon (lat. *diencephalon*); 3 –midbrain (lat. *mesencephalon*); 4 –hindbrain (lat. *rhombencephalon*); 5 –medulla oblongata (lat. *myelencephalon*); 6 –spinal cord (lat. *medulla spinalis*); 7 –pineal gland (lat. *corpus pineale*); 8 –pituitary gland (lat. *hypophysis*); 9 –eyes (lat. *oculus*); 10 –chiasma and optic nerves (lat. *chiasma opticum*). Visualization of structures in the DataViewer software and CTvox software.
(TIF)

**S3 Fig.**  Representative cross-sectional images of the head region of a chick embryo (day 9, embryonic stage HH35), counterstained with 1% PTA for 96 h at 40°C (A): coronal (X-Z), transaxial (X-Y) and sagittal (Z-Y) planes and isosurface 3D renderings of the head region (B). The following structures are marked on the images: 1 –telencephalon (lat. *telencephalon*); 2 –diencephalon (lat. *diencephalon*); 3 –midbrain (lat. *mesencephalon*); 4 –erebellum (lat. *cerebellum*); 5 –fourth ventricle (lat. *ventriculus quartus*); 6 –pons (lat. *pons*); 7 –medulla oblongata (lat. *myelencephalon*); 8 –spinal cord (lat. *medulla spinalis*); 9 –pineal gland (lat. *corpus pineale*); 10 –pituitary gland (lat. *hypophysis*); 11 –eyes (lat. *oculus*); 12 –chiasma and optic nerves (lat. *chiasma opticum*). Visualization of structures in the DataViewer software and CTvox software.
(TIF)

**S4 Fig.**  Representative cross-sectional images of the head region of a chick embryo (day 10, embryonic stage HH36), counterstained with 1% PTA for 96 h at 40°C (A): coronal (X-Z), transaxial (X-Y) and sagittal (Z-Y) planes and isosurface 3D renderings of the head region (B). The following structures are marked on the images: 1 –telencephalon (lat. *telencephalon*); 2 –diencephalon (lat. *diencephalon*); 3 –midbrain (lat. *mesencephalon*); 4 –erebellum (lat. *cerebellum*); 5 –fourth ventricle (lat. *ventriculus quartus*); 6 –pons (lat. *pons*); 7 –medulla oblongata (lat. *myelencephalon*); 8 –spinal cord (lat. *medulla spinalis*); 9 –pineal gland (lat. *corpus pineale*); 10 –pituitary gland (lat. *hypophysis*); 11 –eyes (lat. *oculus*); 12 –chiasma and optic nerves (lat. *chiasma opticum*). Visualization of structures in the DataViewer software and CTvox software.
(PNG)

**S5 Fig.**  Representative cross-sectional images of the head region of a chick embryo (day 12, embryonic stage HH38), counterstained with 1% PTA for 96 h at 40°C (A): coronal (X-Z), transaxial (X-Y) and sagittal (Z-Y) planes and isosurface 3D renderings of the head region (B). The following structures are marked on the images: 1 –telencephalon (lat. *telencephalon*); 2 –diencephalon (lat. *diencephalon*); 3 –midbrain (lat. *mesencephalon*); 4 –erebellum (lat. *cerebellum*); 5 –fourth ventricle (lat. *ventriculus quartus*); 6 –pons (lat. *pons*); 7 –medulla oblongata (lat. *myelencephalon*); 8 –spinal cord (lat. *medulla spinalis*); 9 –pineal gland (lat. *corpus pineale*); 10 –pituitary gland (lat. *hypophysis*); 11 –eyes (lat. *oculus*); 12 –chiasma and optic nerves (lat. *chiasma opticum*). Visualization of structures in the DataViewer software and CTvox software.
(PNG)

**S6 Fig.**  Representative cross-sectional images of the head region of a chick embryo (day 13, embryonic stage HH39), counterstained with 1% PTA for 96 h at 40°C (A): coronal (X-Z), transaxial (X-Y) and sagittal (Z-Y) planes and isosurface 3D renderings of the head region (B).

The following structures are marked on the images: 1 –telencephalon (lat. *telencephalon*); 2 –diencephalon (lat. *diencephalon*); 3 –midbrain (lat. *mesencephalon*); 4 –erebellum (lat. *cerebellum*); 5 –fourth ventricle (lat. *ventriculus quartus*); 6 –pons (lat. *pons*); 7 –medulla oblongata (lat. *myelencephalon*); 8 –spinal cord (lat. *medulla spinalis*); 9 –pineal gland (lat. *corpus pineale*); 10 –pituitary gland (lat. *hypophysis*); 11 –eyes (lat. *oculus*); 12 –chiasma and optic nerves (lat. *chiasma opticum*). Visualization of structures in the DataViewer software and CTvox software.
(PNG)

**S7 Fig.**  Representative cross-sectional images of the head region of a chick embryo (day 15, embryonic stage HH41), counterstained with 1% PTA for 96 h at 40˚C (A): coronal (X-Z), transaxial (X-Y) and sagittal (Z-Y) planes and isosurface 3D renderings of the head region (B). The following structures are marked on the images: 1 –telencephalon (lat. *telencephalon*); 2 –diencephalon (lat. *diencephalon*); 3 –midbrain (lat. *mesencephalon*); 4 –erebellum (lat. *cerebellum*); 5 –fourth ventricle (lat. *ventriculus quartus*); 6 –pons (lat. *pons*); 7 –medulla oblongata (lat. *myelencephalon*); 8 –spinal cord (lat. *medulla spinalis*); 9 –pineal gland (lat. *corpus pineale*); 10 –pituitary gland (lat. *hypophysis*); 11 –eyes (lat. *oculus*); 12 –chiasma and optic nerves (lat. *chiasma opticum*). Visualization of structures in the DataViewer software and CTvox software.
(PNG)

**S8 Fig. Representative horizontal sectional images of the head region of chick embryo (day 4, embryonic stages HH22-24), counterstained with 1% PTA for 24 h at 40˚C.** Scale ruler—1 mm. The following structures are marked on the images: AA–aortic arch artery; Acardv–anterior cardinal vein; Aq–cerebral aqueduct; Daorta–dorsal aorta; Hy–hypothalamus Mesen–mesencephalon; Noto–notochord; Oral–oral cavity; Parch–pharyngeal arch; Som–somite; SpC–spinal cord; Tect–tectum; Tongue–tongue; V3V –ventral third ventricle; Vitr–vitreous humor of eye.
(PNG)

**S9 Fig. Representative horizontal sectional images of the head region of chick embryo (day 5, embryonic stages HH25-27), counterstained with 1% PTA for 24 h at 40˚C.** Scale ruler—1 mm. The following structures are marked on the images: 4V –fourth ventricle; 5Gn–trigeminal ganglion; Acardv–anterior cardinal vein; Aq–cerebral aqueduct; CD–cochlear duct; Daorta–dorsal aorta; DRG–dorsal root ganglion; Fovis–fovea of isthmus; Mesen–mesencephalon; Noto–notochord; Rhomb–rhombencephalon; Som–somite; SpC–spinal cord; Tect–tectum.
(PNG)

**S10 Fig. Representative horizontal sectional images of the head region of chick embryo (day 6, embryonic stages HH28-29), counterstained with 1% PTA for 24 h at 40˚C.** Scale ruler—1 mm. The following structures are marked on the images: Acardv–anterior cardinal vein; D3V –dorsal third ventricle; Dien–diencephalon; Eso–esophagus; Lens–lens; LV–lateral ventricle; Noto–notochord; OptRe–optic recess of 3[rd] ventricle. Oral–oral cavity; Os–optic stalk; SpC–spinal cord; Telen–telencephalon; Tongue–tongue; Trachea–trachea; V3V –ventral third ventricle; Vitr–vitreous humor of eye.
(PNG)

**S11 Fig. Representative horizontal sectional images of the head region of chick embryo (day 7, embryonic stages HH30-32), counterstained with 1% PTA for 24 h at 40˚C.** Scale ruler—1 mm. The following structures are marked on the images: 5Gn–trigeminal ganglion;

ELD–endolymphatic duct; GG–glossopharyngeal ganglion; Hy–hypothalamus; Ivf–interventricular foramen of Monro; LV–lateral ventricle; Myelen–myelencephalon; Sacc–saccule of inner ear; SpC–spinal cord; SSCD–superior semicircular duct; Telen–telencephalon; Utr–utricle of inner ear; V3V –ventral third ventricle; Vitr–vitreous humor of eye.
(PNG)

**S12 Fig. Representative horizontal sectional images of the head region of chick embryo (day 8, embryonic stages HH33-34), counterstained with 1% PTA for 24 h at 40˚C.** Scale ruler—1 mm. The following structures are marked on the images: 5Gn–trigeminal ganglion; Chp–choroid plexus; Dien–diencephalon; Hy–hypothalamus; LV–lateral ventricle; Pons–pons; Sacc–saccule of inner ear; SpC–spinal cord; V3V –ventral third ventricle; Vitr–vitreous humor of eye.
(PNG)

**S13 Fig. Representative horizontal sectional images of the head region of chick embryo (day 9, embryonic stage HH35), counterstained with 1% PTA for 96 h at 40˚C.** Scale ruler—1 mm. The following structures are marked on the images: 4V –fourth ventricle; Cb–erebellum; Chiasma–optic chiasma; Hy–hypothalamus; Pons–pons; V3V –ventral third ventricle; Vitr–vitreous humor of eye.
(PNG)

**S14 Fig. Representative horizontal sectional images of the head region of chick embryo (day 10, embryonic stage HH36), counterstained with 1% PTA for 96 h at 40˚C.** Scale ruler—2 mm. The following structures are marked on the images: Coch–cochlea; Dien–diencephalon; GG–glossopharyngeal ganglion; Hy–hypothalamus; Mesen–mesencephalon; OB–olfactory bulb; OV–olfactory ventricle; Pons–pons; PSCD–posterior semicircular duct; Sacc–saccule of inner ear; SSCD–superior semicircular duct; Telen–telencephalon; Utr–utricle of inner ear; V3V –ventral third ventricle; Vitr–vitreous humor of eye.
(TIF)

**S15 Fig. Representative horizontal sectional images of the head region of chick embryo (day 11, embryonic stage HH37), counterstained with 1% PTA for 96 h at 40˚C.** Scale ruler—2 mm. The following structures are marked on the images: 4V –fourth ventricle; Aq–cerebral aqueduct; D3V –dorsal third ventricle; LRAq–lateral recess of the cerebral aqueduct; Mesen–mesencephalon; Myelen–myelencephalon; Pi–pineal gland; Pons–pons; SpC–spinal cord; Telen–telencephalon.
(TIF)

**S16 Fig. Representative horizontal sectional images of the head region of chick embryo (day 12, embryonic stage HH38), counterstained with 1% PTA for 96 h at 40˚C.** Scale ruler—2 mm. The following structures are marked on the images: 4V –fourth ventricle; Cb–erebellum; Chp–choroid plexus; Dien–diencephalon; LRAq–lateral recess of the cerebral aqueduct; LV–lateral ventricle; Mesen–mesencephalon; Myelen–myelencephalon; PSe–pallial septum; Sacc–saccule of inner ear; Sm/ohb–stria medullaris talami/olfactohabenular tract; SSCD–superior semicircular duct; Telen–telencephalon; Utr–utricle of inner ear; Vitr–vitreous humor of eye.
(TIF)

**S17 Fig. Representative horizontal sectional images of the head region of chick embryo (day 13, embryonic stage HH39), counterstained with 1% PTA for 96 h at 40˚C.** Scale ruler—2 mm. The following structures are marked on the images: 4V –fourth ventricle; Cb–erebellum; Chp–choroid plexus; Ivf–interventricular foramen of Monro; LRAq–lateral recess of the

cerebral aqueduct; LV–lateral ventricle; Mesen–mesencephalon; Pi–pineal gland; PSCD–posterior semicircular duct; PSe–pallial septum; Sacc–saccule of inner ear; SSCD–superior semicircular duct; Telen–telencephalon; Utr–utricle of inner ear.
(PNG)

**S18 Fig. Representative horizontal sectional images of the head region of chick embryo (day 14, embryonic stage HH40), counterstained with 1% PTA for 96 h at 40˚C.** Scale ruler —2 mm. The following structures are marked on the images: CCrus–common crus of semicircular duct; Dien–diencephalon; LRAq–lateral recess of the cerebral aqueduct; LV–lateral ventricle; Mesen–mesencephalon; Myelen–myelencephalon; Pons–pons; PSCD–posterior semicircular duct; Sacc–saccule of inner ear; Telen–telencephalon; Utr–utricle of inner ear; Vitr–vitreous humor of eye.
(PNG)

**S19 Fig. Representative horizontal sectional images of the head region of chick embryo (day 15, embryonic stage HH41), counterstained with 1% PTA for 96 h at 40˚C.** Scale ruler —2 mm. The following structures are marked on the images: Chiasma–optic chiasma; Dien–diencephalon; Hy–hypothalamus; Mesen–mesencephalon; Myelen–myelencephalon; Pons–pons; Sacc–saccule of inner ear; SSCD–superior semicircular duct; Utr–utricle of inner ear; V3V –ventral third ventricle; Vitr–vitreous humor of eye.
(PNG)

**S1 Raw data.**
(ZIP)

## Author Contributions

**Conceptualization:** Mohammad Ali Shariati, Faisal Aqlan.

**Data curation:** Andrey Nagdalian.

**Formal analysis:** Andrey Nagdalian.

**Funding acquisition:** Igor Rzhepakovsky, Lyudmila Timchenko, Ammar Al-Farga.

**Investigation:** Svetlana Avanesyan.

**Methodology:** Svetlana Avanesyan.

**Resources:** Magomed Shakhbanov, Lyudmila Timchenko, Mohammad Ali Shariati, Andrey Likhovid.

**Software:** Magomed Shakhbanov, Marina Sizonenko, Faisal Aqlan.

**Supervision:** Andrey Likhovid.

**Writing – original draft:** Igor Rzhepakovsky, Sergey Piskov.

**Writing – review & editing:** Sergey Piskov, Marina Sizonenko.

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
