## [Decision Letter · Decision Letter 0]

23 Jul 2024

PONE-D-24-27518Expanding understanding of chick embryo’s nervous system development at HH22-HH41 embryonic stages using X-ray microcomputed tomographyPLOS ONE

Dear Dr. Aqlan,

Thank you for submitting your manuscript to PLOS ONE. After careful consideration, we feel that it has merit but does not fully meet PLOS ONE’s publication criteria as it currently stands. Therefore, we invite you to submit a revised version of the manuscript that addresses the points raised during the review process. 

We look forward to receiving your revised manuscript.

Kind regards,

Mani Alikhani, DDS,MS, PhD

Academic Editor

PLOS ONE

Journal Requirements:

6. Please amend the manuscript submission data (via Edit Submission) to include author "Ammar AL-Farga".

7. Your ethics statement should only appear in the Methods section of your manuscript. If your ethics statement is written in any section besides the Methods, please delete it from any other section. 

8. Please include a separate caption for each figure in your manuscript.

**Additional Editor Comments:**

Reviewers mostly request more clarification on method and materials and method of writings.

Reviewers' comments:

Reviewer's Responses to Questions

**Comments to the Author**

1. Is the manuscript technically sound, and do the data support the conclusions?

Reviewer #1: Yes

Reviewer #2: Partly

2. Has the statistical analysis been performed appropriately and rigorously? 

Reviewer #1: Yes

Reviewer #2: Yes

3. Have the authors made all data underlying the findings in their manuscript fully available?

Reviewer #1: Yes

Reviewer #2: Yes

4. Is the manuscript presented in an intelligible fashion and written in standard English?

Reviewer #1: Yes

Reviewer #2: No

5. Review Comments to the Author

Reviewer #1: The authors provide a valuable tool for researchers interested in CNS development. The techniques used provided clear histological and skeletal images of normal chicken craniofacial development and the authors plan to make the database available to researchers.

Reviewer #2: In this study, Rzhepakovsky and co-workers aim to present a simple method for obtaining very detailed quantitative sets of 2D and 3D high-resolution images of HH22-HH41 chick embryo stages using X-ray microcomputed tomography. These images constitute the basis for a brain chick atlas showing morphological details of nervous system development and dynamics.

The development of a comprehensive methodology for obtaining very detailed quantitative sets of 2D and 3D high-resolution images is important and worth pursuing. However, the present report is not complete since many methodological details are missing. The manuscript should be revised to provide a detailed account in the methods section.

Abstract and Methods

The abstract requires adjustments. Authors claim that “The results obtained demonstrate that μCT is an effective method of quantitative visualization of the CE NS at embryotoxicity and teratogenicity assessment”. This is an overinterpretation of the results since they are only derived from normal tissues. The report is mainly technical report. Authors must tone down their conclusions.

Many methodological points are not clear. Some (although not all of them) are:

How was each egg screened daily to confirm viability?

How were chick embryos from stages HH35-HH41 fixed and dehydrated? The staining protocol needs to be clarified.

Is the contrast staining reagent (1% phosphotungstic acid) the same as the radiopaque staining reagent? Detail staining steps for each HH stage are required.

A reference for the hematoxylin and eosin staining method is missing. The method should also be described briefly.

To clarify the methods a figure describing the staining procedure and a figure for the X-ray microcomputed tomography procedure would be very useful.

Embryos were scanned in test tubes. Do you have any specifications for the tubes?

Why were samples scanned in 70% ethanol?

Segmentation and quantification were carried out according to the recommendations of Kim et al. [42]. A brief description of these recommendations should be included.

Which post hoc test was used?

What do authors mean by "the most characteristic representative materials were used"? This needs clarification.

Results and Discussion section

Authors mention that microcomputed tomography is the most advantageous method over magnetic resonance imaging, optical coherence tomography, high-frequency ultrasound imaging, and mesoscale selective planar illumination microscopy (mesoSPIM). A brief comparison of the advantages and disadvantages of each of the methods must be included to support authors’ claim.

Besides phosphotungstic acid, other substances are used to contrast tissues: osmium tetroxide and iodine-based staining. Why did authors choose phosphotungstic acid? A discussion of the different substances used for contrast tissues should be included.

The main subject of this paper is the description of the figures. The authors must explain the panels of the figures in greater detail.

Figures 2-6 each have five panels, but only three are labeled and described in the figure legend. This is confusing. Figures should be organized so that all panels are entirely described.

In figures 2-6, the lines indicating orientation for each figure panel are not properly labeled or described. This makes interpretation of figures very difficult and confusing. Including a 3D representation as presented in your previous work, Rzhepakovsky, et al, Applied Sciences 13 (2023) 10642, would be very helpful to the reader.

Images and information from Supplementary Figures 1, 3, 5, 8, and 11 are already presented in Figures 2, 3, 4, 5, and 6, respectively. Same images should not be repeated in supplementary data. A careful selection of images and proper description of them in both figures and supplementary figures is required to make this manuscript clear.

Language

The manuscript contains multiple spelling and grammatical errors. A careful revision of the manuscript is needed for the proper use of the English language. Several nouns and pronouns are missing and there is no correlation among nouns and verbs.

The words “staining” and “stain” are used interchangeably. They do not mean the same.

Similarly, the terms “chick” and “chicken embryo” are mixed throughout the manuscript.

The word “mortified” does not mean to kill.

Also, the words “labeled” and “marked” not always mean the same. It would be helpful to use only one term consistently throughout the manuscript.

Please revise carefully.

Minor considerations

References 11 and 12 explain neural tube defects using chick embryos, but not in humans. Therefore, these references are not properly cited. Please revise the manuscript carefully for the proper references

When referring to 3D visualization methods, the species used (axolotl, zebrafish, Xenopus, mice, etc.) should be included.

Table 2 contains data, not results. This should be revised carefully.

The sentence, "μCT is the most advantageous method in this regard because of relatively high speed of scanning, diagnostic accuracy, high resolution, and ability to visualize the entire internal 3D structure of the object with its complete preservation for other types of research" is very confusing. Are authors comparing methods with types of research? This should be revised and explained better.

6. PLOS authors have the option to publish the peer review history of their article (what does this mean?). If published, this will include your full peer review and any attached files.

Reviewer #1: No

Reviewer #2: **Yes: **Eileen Uribe-Querol

---

## [Author Response · Author response to Decision Letter 0]

26 Aug 2024

We are grateful to the Editor and Reviewers for their positive evaluation and for the time devoted to review our manuscript. All comments were useful and pleased us with the high level of understanding of the topic. We have addressed all recommendations as requested. All changes were marked by green (revisions regarding comments and recommendations) and red (English correction). Please see the point-by-point response below.

Editor:

Please include your amended statements within your cover letter; we will change the online submission form on your behalf. Response: Done

4. Please provide a complete Data Availability Statement in the submission form, ensuring you include all necessary access information or a reason for why you are unable to make your data freely accessible. If your research concerns only data provided within your submission, please write "All data are in the manuscript and/or supporting information files" as your Data Availability Statement. Response: Done

5. PLOS requires an ORCID iD for the corresponding author in Editorial Manager on papers submitted after December 6th, 2016. Please ensure that you have an ORCID iD and that it is validated in Editorial Manager. To do this, go to ‘Update my Information’ (in the upper left-hand corner of the main menu), and click on the Fetch/Validate link next to the ORCID field. This will take you to the ORCID site and allow you to create a new iD or authenticate a pre-existing iD in Editorial Manager. Please see the following video for instructions on linking an ORCID iD to your Editorial Manager account: https://www.youtube.com/watch?v=_xcclfuvtxQ Response: Done

6. Please amend the manuscript submission data (via Edit Submission) to include author "Ammar AL-Farga". 

Response: Done

7. Your ethics statement should only appear in the Methods section of your manuscript. If your ethics statement is written in any section besides the Methods, please delete it from any other section. 

8. Please include a separate caption for each figure in your manuscript. 

Response: There were all captions in the manuscript

Response: added 

10. Please review your reference list to ensure that it is complete and correct. If you have cited papers that have been retracted, please include the rationale for doing so in the manuscript text, or remove these references and replace them with relevant current references. Any changes to the reference list should be mentioned in the rebuttal letter that accompanies your revised manuscript. If you need to cite a retracted article, indicate the article’s retracted status in the References list and also include a citation and full reference for the retraction notice. Response: Done

Additional Editor Comments:

Reviewers mostly request more clarification on method and materials and method of writings.

Reviewers' comments:

Reviewer's Responses to Questions

Comments to the Author

1. Is the manuscript technically sound, and do the data support the conclusions?

Reviewer #1: Yes

Reviewer #2: Partly

2. Has the statistical analysis been performed appropriately and rigorously?

Reviewer #1: Yes

Reviewer #2: Yes

3. Have the authors made all data underlying the findings in their manuscript fully available?

Reviewer #1: Yes

Reviewer #2: Yes

4. Is the manuscript presented in an intelligible fashion and written in standard English?

Reviewer #1: Yes

Reviewer #2: No 

5. Review Comments to the Author

Reviewer #1: The authors provide a valuable tool for researchers interested in CNS development. The techniques used provided clear histological and skeletal images of normal chicken craniofacial development and the authors plan to make the database available to researchers.

Reviewer #2: In this study, Rzhepakovsky and co-workers aim to present a simple method for obtaining very detailed quantitative sets of 2D and 3D high-resolution images of HH22-HH41 chick embryo stages using X-ray microcomputed tomography. These images constitute the basis for a brain chick atlas showing morphological details of nervous system development and dynamics.

The development of a comprehensive methodology for obtaining very detailed quantitative sets of 2D and 3D high-resolution images is important and worth pursuing. However, the present report is not complete since many methodological details are missing. The manuscript should be revised to provide a detailed account in the methods section.

Abstract and Methods

The abstract requires adjustments. Authors claim that “The results obtained demonstrate that μCT is an effective method of quantitative visualization of the CE NS at embryotoxicity and teratogenicity assessment”. This is an overinterpretation of the results since they are only derived from normal tissues. The report is mainly technical report. Authors must tone down their conclusions.

Response: Thank you for recommendation! We revised the last sentence of the Abstract and mentioned that the data obtained open up new opportunities for modern embryology, teratology, pharmacology and toxicology 

Many methodological points are not clear. Some (although not all of them) are:

How was each egg screened daily to confirm viability?

Response: Each egg was screened daily to confirm viability and level of embryos development was monitored using the PKYA-10 ovoscope (Russia). The correspoindong information was added to subsection 2.2.

How were chick embryos from stages HH35-HH41 fixed and dehydrated? The staining protocol needs to be clarified.

Response: The CE (9-15 day, HH35-HH41) fixed in a 10% buffered formalin solution for 96 hours were washed under running water for 24 hours, dehydrated in replaceable portions of ethanol 30% (2 hours), 50% (2 hours), 70% (12 hours) and placed in solution of radiopaque stain 1% PTA at 1:20 (V of CE to V of solution) for 96 hours. The corresponding information was added to subsection 2.3.

Is the contrast staining reagent (1% phosphotungstic acid) the same as the radiopaque staining reagent? Detail staining steps for each HH stage are required.

Response: Thank you for the comment. Yes, 1% phosphotungstic acid for staining at all stages was the same. Only the volume used differed, which depended on the volume of the embryo and the exposure time. For instance, at stages 4-8 day, HH22-HH34 – 24 hours, and at stages 9-15 day, HH35-HH41 – 96 hours, as we indicated in the methodology

A reference for the hematoxylin and eosin staining method is missing. The method should also be described briefly.

Response: The histological sections were stained with hematoxylin and eosin in accordance with generally accepted protocols (Wick, 2019). Briefly, the histological sections were dewaxed by incubation in xylene (2 cycles), dehydrated in ethanol at concentrations of 95%, 80% and 70%, and washed with distilled water. Then, the histological sections were incubated in hematoxylin solution (3 minutes), washed in tap water, incubated in 1% aqueous eosin solution (5 minutes) and washed with distilled water. After removing the water spills, the histological sections were incubated in 96% ethanol and xylene and closed in a mounting medium Vitrogel (Biovitrum, St. Petersburg, Russia). 

The corresponding information was added to subsection 2.5.

Reference: Wick, M.R. (2019). The hematoxylin and eosin stain in anatomic pathology—An often-neglected focus of quality assurance in the laboratory. Seminars in Diagnostic Pathology 36, 303–311. https://doi.org/10.1053/j.semdp.2019.06.003

To clarify the methods a figure describing the staining procedure and a figure for the X-ray microcomputed tomography procedure would be very useful.

Response: Thank you for the comment. We would like to share here some photos from the experiment. However, we believe, it is better not to overload the manuscript with additional figures as we already put many figures in Supplementary. Hope for understanding. 

Embryos were scanned in test tubes. Do you have any specifications for the tubes?

Why were samples scanned in 70% ethanol?

Response: For µCT, Eppendorf Safe-Lock Tubes 2 mL colorless (polypropylene) were used for 4-7 days CE (HH22-HH32), Servicebio Centrifuge Tubes 15 mL colorless (polypropylene) were used for 8-12 days CE (HH33-HH38) and Servicebio Centrifuge Tubes BioBased 50 mL colorless (polypropylene) were used for 13-15 days CE (HH39-HH41). 

The corresponding information was added to subsection 2.4

The use of 70% ethanol solution is justified by low radiopacity, which makes it possible to clearly visualize even the lowest contrast parts of the object. This was checked and proved by us in number of previous works as well as confirmed by other researchers. 

References: 

Handschuh, S., and Glösmann, M. (2022). Mouse embryo phenotyping using X-ray microCT. Front. Cell Dev. Biol. 10. https://doi.org/10.3389/fcell.2022.949184

Metscher, B. (2020). A simple nuclear contrast staining method for microCT‐based 3D histology using lead(II) acetate. Journal of Anatomy 238, 1036–1041. https://doi.org/10.1111/joa.13351

An interesting comparative example was obtained by Handschuh and Glösmann (2022), see figure below. 

It is worth noting, that air was not suitable for CE because of natural drying processes. 

Segmentation and quantification were carried out according to the recommendations of Kim et al. [42]. A brief description of these recommendations should be included.

Response: 

Volume segmentation of 3D images was performed using the algorithm recommended by Bruker-microCT (Kontich, Belgium). A consistent allocation of areas of interest was carried out. The resulting structures were saved as separate volumes, or most of the quantitative calculations present in this article, the organ or region of interest was segmented by creating negative space (i.e., setting pixel values in that region to zero contrast) and propagating that negative space through the Z plane. In this way, the only one tissue type was captured which allowed to independently quantify tissue or fluid volume in a particular organ. Multiple labeled volumes were derived from each CE and quantified in the same way. Data from at least five embryos were used for each region/tissue.

The corresponding information was added to subsection 2.4.

Skyscan 1176 (software platform Bruker, Kontich, Belgium) running on a Windows 7 Professional (Microsoft Corp., Redmond, WA, USA) workstation with 32 Gb of RAM and an Nvidia Quadro K 4000 graphics card (Nvidia Corp., Santa Clara, CA, USA) was used for µCT data processing.

The corresponding information was added to subsection 2.6.

Which post hoc test was used?

Response: The assessment of individual differences in the samples was carried out using statistical analysis using ANOVA, followed by Tukey post hoc testing using p < 0.05 as a significance threshold.

The corresponding information was added to subsection 2.6.

What do authors mean by "the most characteristic representative materials were used"? This needs clarification.

Response: The selection of materials consisted in excluding from research digital material containing mechanical and digital defects that could appear as a result of staining or scanning.

The corresponding information was added to subsection 2.6.

Results and Discussion section

Authors mention that microcomputed tomography is the most advantageous method over magnetic resonance imaging, optical coherence tomography, high-frequency ultrasound imaging, and mesoscale selective planar illumination microscopy (mesoSPIM). A brief comparison of the advantages and disadvantages of each of the methods must be included to support authors’ claim.

Response: Thank you for recommendation. We expanded this part. 

Besides phosphotungstic acid, other substances are used to contrast tissues: osmium tetroxide and iodine-based staining. Why did authors choose phosphotungstic acid? A discussion of the different substances used for contrast tissues should be included.

Response: Previously, we conducted a theoretical and practical analysis of the most accessible contrasts used in embryology, including osmium tetroxide, silver proteinate, eosin and iodine-based staining. Osmium tetroxide, despite its fairly widespread use in electron microscopy and microCT, is an extremely toxic substance and based on the results of Kim et al., Lesciotto et al., Metscher and Handschuh we recognized its insufficient effectiveness for use in large-scale research. The use of silver proteinate, eosin and Lugol's solution (I2KI) in various concentrations during the CE contrast did not make it possible to obtain microCT results providing reliable microstructural analysis. Silver proteinate and eosin (1% and 5%) staining, in addition to poor differential contrast, had layering of stain on the external tissues of CE, which in-creased its natural size and deformed the contours. It was observed, that I2KI staining significantly reduces distance between CE organs and decreases the level of differential contrast and the clarity of the boundaries between internal organs, even with an increased value of the X-ray density of the tissue. Our results showed, that the mos

---

## [Editor Report · Decision Letter 1]

2 Sep 2024

Expanding understanding of chick embryo’s nervous system development at HH22-HH41 embryonic stages using X-ray microcomputed tomography

PONE-D-24-27518R1

Dear Dr. Aqlan,

We’re pleased to inform you that your manuscript has been judged scientifically suitable for publication and will be formally accepted for publication once it meets all outstanding technical requirements.

Kind regards,

Mani Alikhani, DDS,MS, PhD

Academic Editor

PLOS ONE
---

## [Editor Report · Acceptance letter]

5 Nov 2024

PONE-D-24-27518R1 

PLOS ONE

Dear Dr. Aqlan, 

I'm pleased to inform you that your manuscript has been deemed suitable for publication in PLOS ONE. Congratulations! Your manuscript is now being handed over to our production team.

Kind regards, 

on behalf of

Dr. Mani Alikhani 

Academic Editor

PLOS ONE